# Relationship between Facial Color Changes and Psychological Problems Associated with Lower Back Pain

**DOI:** 10.3390/medicina58101471

**Published:** 2022-10-17

**Authors:** Shinji Tanishima, Yasunori Kotani, Chikako Takeda, Tokumitsu Mihara, Shinya Ogawa, Akira Matsubara, Takashi Goto, Takahiro Hirayama, Hideki Hashizume, Junichiro Arai, Daichi Mukunoki, Hideki Nagashima

**Affiliations:** 1Division of Orthopedic Surgery, Department of Sensory and Motor Organs, School of Medicine, Tottori University Faculty of Medicine, Tottori 683-8504, Japan; 2Department of Social and Human Sciences Institute for Liberal Arts Tokyo Institute of Technology Japan, School of Environment and Society, Tokyo 152-8552, Japan; 3Department of Orthopedic Surgery, Masuda Red Cross Hospital, Shimane 698-8501, Japan; 4Defense Systems Division, Daikin Industries, LTD., Osaka 530-8323, Japan; 5Technology and Innovation Center, Daikin Industries, LTD., Osaka 566-8585, Japan; 6Department of Rehabilitation, Yowa Hospital, Tottori 683-0841, Japan

**Keywords:** non-contacting sensor, lower back pain, brain activity, emotion, pain catastrophizing

## Abstract

*Background and Objectives*: The aim of this study was to determine whether a non-contact sensor that detects complexion changes can be used to assess the psychological state of patients with chronic lower back pain (LBP). *Materials and Methods*: Twenty-six patients with LBP (LBP group; mean age = 68.0 ± 13.9 years) and 18 control subjects without LBP (control group; mean age = 60.8 ± 16.1 years) were included in the study. All the subjects in the two groups wore headphones when asked LBP-related and LBP-unrelated questions. During questioning, the facial image of the subjects was captured using a video camera, and the complexion of the subjects was converted into red, green, and blue (RGB) values. RGB correlation coefficients (RGBCCs; range: 0–1) represent the difference in complexion between LBP-related and LBP-unrelated questions. A high RGBCC indicates that the brain is more activated by LBP-related questions than by LBP-unrelated questions. We also noted the scores of subjects on the Numerical Rating Scale (NRS), Japanese Orthopedic Association Back Pain Evaluation Questionnaire (JOABPEQ), Pain Catastrophizing Scale (PCS), and Hospital Anxiety and Depression Scale (HADS). *Results*: There were no significant differences in RGBCC between the control and LBP groups (0.64 versus 0.56, *p* = 0.08). In the LBP group, no correlation was observed between RGBCC and each examination item of NRS, JOABPEQ, and HADS. In contrast, a correlation was observed between RGBCC and the rumination subscale of PCS in the LBP group (Spearman’s rank correlation coefficient = 0.40, *p* = 0.04). *Conclusions*: The complexion of patients with catastrophic thinking changes when the patients are asked LBP-related questions.

## 1. Introduction

It is a well-known fact that chronic pain affects the mental state of patients [1]. Lower back pain (LBP) accounts for most chronic pain cases. According to the 2009 National Health Interview Survey in the USA, 28% of adults have persistent LBP [2], and it has often been reported that depression and history of LBP are associated with poor recovery from LBP [3]. Pain catastrophizing affects the perceived intensity of pain, duration of pain, physical function, and treatment outcomes [4,5]. Functional magnetic resonance imaging (fMRI) has shown that the sensory discrimination system of the brain is active in patients with acute LBP, while the emotional and cognitive network is dominantly active in patients with chronic LBP [6,7]. It has been shown that patients with chronic LBP have altered brain responses. Hence, given that chronic LBP also affects the mental state of patients, comprehensive treatment should also address the psychological problem of patients [8]. Thus, it is essential that the mental state of patients with chronic LBP is evaluated. Physicians often use questionnaires such as Pain Catastrophizing Scale (PCS), Hospital Anxiety and Depression Scale (HADS), and Fear-Avoidance Beliefs Questionnaire to detect mental problems [9,10,11], and they use Japanese Orthopedic Association Back Pain Evaluation Questionnaire (JOABPEQ) to assess mental status related to LBP [12]. In addition to these questionnaires, various procedures used to investigate brain activity can be used to detect mental problems. These procedures include positron emission tomography, single-photon emission tomography, electroencephalography, fMRI, and near-infrared spectroscopy [13,14,15,16]. These modalities measure brain activity directly by analyzing images. The use of these methods for the investigation of mental disorders was reported in some studies [17,18]. Although these procedures are better for the assessment of mental status than questionnaires, it is difficult to use them in clinical practice. This difficulty limits the use of these procedures for the evaluation of patients with LBP. A promising approach would be to find new procedures for brain activity measurement that allow for better and easier assessment of mental stress in patients with LBP. Fuji et al. reported that erythema index changes on facial images captured with a video camera can be used to assess discomfort associated with colonoscopy [19]. Indeed, the erythema index depends on facial redness, which reflects facial hemodynamics. Moreover, in response to high brain temperature due to increased brain activity, blood flow from the face to the crown is increased to cool the brain, and the cooling occurs by heat exchange between blood vessels and nerve plexuses. In the study by White et al., it was reported that facial blood flow increases due to cooling of the brain during brain activity [20]. Therefore, we hypothesized that the erythema index can be used to assess mental stress in patients with chronic LBP. The aim of this study was to determine whether a non-contact brain activity detection sensor can be used to detect the facial color of patients with chronic LBP and to investigate the association between facial color, evaluation scores of LBP, and psychological state of patients with LBP.

## 2. Materials and Methods

This study was a single-center prospective observational study. Patients with chronic LBP who visited our hospital between 2016 and 2019 were recruited for this study.

The inclusion criteria were patients who had chronic LBP with NRS 5 or higher lasting for at least three months. The exclusion criteria were infection, tumor, mental disorder, and hearing impairment. We posted flyers about this study in our hospital to recruit volunteers. These volunteers included healthy subjects and patients. No volunteer who was a patient had a diagnosis of a disease related to LBP.

The subjects were given headphones and were asked LBP-related questions and general LBP-unrelated questions (Figure 1).

While the subjects listened to both types of questions (Table 1), their facial images were recorded using a color video camera at 30 frames per second, 1920 × 1080 pixels, H.264 (MPEG-4 AVC) compression format, and 15 Mbps bit rate [21].

The rectangular area on the participants’ facial images defined by the outer edges of both eyebrows and the base of the nose was designated as the region of interest (ROI) for analysis (Figure 2). 

We extracted the red, green, and blue (RGB) values (8-bit color depth) contained in each pixel in the ROI of all image frames. Changes in RGB data when the subjects heard each question were quantified using a non-contact activity sensing system [22]. To determine the erythema index (*a**), the following equation used to convert an RGB value to erythema index was applied (CIE-L**a*b** color system) [23]:

α*** = 500{${\left(\frac{X}{Xm}\right)}^{\frac{1}{3}}-{\left(\frac{Y}{Ym}\right)}^{\frac{1}{3}}$}, where X = (0.4124 × R) + (0.3576 × G) + (0.1805 × B), Y = (0.2126 × 0.7152 × G) + (0.0722 × B), X_m_ = 98.071, Y_m_ = 100.0, and RGB value range = 0–255. After converting RGB data to erythema index, we created a dataset of erythema index (the number of pixels of ROI by the number of frames). The dataset was subjected to singular value decomposition (SVD) to extract temporal components from the dataset. The extracted components were band-pass-filtered at a cut-off frequency of 0.004–0.05 Hz. We calculated a reference wave to determine whether the extracted components correlate with LBP-related questions. The reference wave consisted of 1 and 0 values; the time interval of LBP-related questions was set as 1 and the time interval of general LBP-unrelated questions was set as 0. Thus, the reference wave was a rectangular wave of 1s (LBP-related questions) and 0s (general LBP-unrelated questions). Correlation analyses were used to determine the correlation coefficients between each component extracted using SVD and the reference wave. The component with the highest correlation coefficient value was selected as a component of interest. The waveform of the selected component was depicted, and its correlation coefficient was defined as RGB correlation coefficient (RGBCC). The range of RGBCC was 0 (no brain activity) to 1 (maximum brain activity). The data processing flow chart is shown in Figure 3. 

These processes were performed based on a previous study that used the same system [19]. Figure 4 shows typical waves obtained during the hearing of questions; the blue wave indicates the waveform of a selected component before band-pass filtering, the orange wave is the component waveform after band-pass filtering to remove noise and drift, and the waveform of the reference wave is shown in red.

### 2.1. Questionnaires

Below are the questionnaires administered to patients with LBP (LBP group).

(1) Numerical Rating Scale (NRS).

The patients were asked to rate their pain intensity on an 11-point Likert scale (0 = no pain, 10 = worst pain imaginable) [24].

(2) Oswestry Disability Index (ODI).

This is a 10-section questionnaire used to assess the level of pain and the interference of pain with physical activities such as sleeping, self-care, sex life, social life, and traveling [25]. For each question, the subjects were asked to select one of the six possible responses (scores from 0 to 5) that best describe their situation. The scores for each section were added, divided by the total possible score (50, if all sections are completed), and shown as percentage score (0% = no disability, 100% = maximum disability). In this study, we used nine questions, excluding the sex life section.

(3) Japanese Orthopedic Association Back Pain Evaluation Questionnaire (JOABPEQ).

This questionnaire is used in Japan to assess LBP and lumbar spinal disease. It has the following five domains: pain-related disorder, lumbar spine dysfunction, gait disturbance, social life disturbance, and psychological disorder [12].

(4) Pain Catastrophizing Scale.

We assessed pain catastrophizing using the PCS. This scale evaluates catastrophic thinking about pain and consists of 13 items that describe the individual’s specific beliefs about their pain [11]. For each question, the subjects selected the most appropriate answer on a Likert scale (0 = not at all, 4 = all the time). All 13 scores were summed up to obtain the final score (range: 0–52). In addition to an overall score, the PCS also provides subscale scores for rumination, magnification, and helplessness.

(5) Hospital Anxiety and Depression Scale.

We used the 14-item HADS [9] to measure anxiety and depression in the study participants. Seven items were related to anxiety, and seven items were related to depression. Each item had a score of 0 to 3, and the higher the score, the greater the depression (HADS-depression) or anxiety (HADS-anxiety).

### 2.2. Statistical Analysis

All the data were expressed as mean ± standard deviation. The subjects were divided into a control group and an LBP group. Differences in RGBCC between the control group and the LBP group were analyzed using Mann–Whitney *U* tests. Correlations between RGBCC and various scores (ODI, JOABPEQ, PCS, and HADS) in the LBP group were evaluated using *Pearson’s* product–moment correlation coefficient. A two-tailed *p* value < 0.05 was considered statistically significant. The data were analyzed using StatMate for Windows (version 4.01; ATMS Corporation, Tokyo, Japan). All the methods used for analysis were in accordance with relevant guidelines and regulations.

## 3. Results

### 3.1. Sample Characteristics of Chronic Lower Back Pain Patients

Twenty-six patients (nine men and seventeen women) were randomly enrolled in the study, and their mean age was 68.0 ± 13.9 years (range: 26–84 years). Chronic LBP was defined as having LBP for at least three months. Diagnosis of LBP included lumbar spinal stenosis (13 patients), spondylosis (2 patients), lumbar vertebral fracture (2 patients), lumbar kyphoscoliosis (2 patients), lumbar spondylosis deformans (2 patients), sacroiliac arthritis (2 patients), and lumbar disc herniation (3 patients). The mean disease period was 43.5 ± 35.4 months (Table 2).

### 3.2. Sample Characteristics of Control Subjects

Eighteen subjects (12 men and 6 women; mean age = 60.8 ± 16.1 years; age range: 32–89 years) were enrolled as control subjects (control group). These volunteers included healthy subjects and patients. Of the volunteers who were patients, two women had knee osteoarthritis, one man had shoulder osteoarthritis, two men had cervical spondylotic myelopathy, one man had rotator cuff injury, two women had hip osteoarthritis, one man had arteriosclerosis obliterans, and one woman had a bruise on the lower leg. The other eight volunteers (seven men and one woman) were healthy subjects (Table 3).

### 3.3. Comparison of RGBCC between the LBP and Control Groups

Figure 5 shows the RGBCC when hearing LBP-related questions in the LBP and control groups. The mean RGBCC was 0.64 ± 0.11 in the LBP group and 0.56 ± 0.14 in the control group. Although the difference in RGBCC between the two groups was not significant, RGBCC was higher in the LBP group than in the control group (*p* = 0.08).

### 3.4. Correlations of RGBCC with ODI and NRS Score in the LBP Group

Figure 6 shows the correlations of RGBCC with ODI and NRS score in the LBP group. There was no correlation between RGBCC and ODI (Spearman’s rank correlation coefficient (rs) = 0.11, *p* = 0.57) or between RGBCC and NRS score (rs = 0.14, *p* = 0.47).

### 3.5. Correlation between RGBCC and JOABPEQ Score in the LBP Group

Figure 7 shows the correlation of RGBCC with the score of each domain of JOABPEQ (pain-related disorders, lumbar function, gait disturbance, social life disorders, and mental disorders). There were no correlations between RGBCC and the score of any domain of JOABPEQ: pain-related disorders (rs = 0.02, *p* = 0.91), lumbar function (rs = 0.004, *p* = 0.98), gait disturbance (rs = −0.06, *p* = 0.70), social life disorders (rs = −0.161, *p* = 0.41), and mental disorders (rs = −0.25, *p* = 0.20).

### 3.6. Correlation between RGBCC and PCS Score in the LBP Group

Figure 8 shows the correlation of RGBCC with the score of each subscale of PCS and with the total PCS score. The total PCS score did not correlate with RGBCC (rs = 0.24, *p* = 0.24). Regarding the subscales, rumination had a significant correlation with RGBCC (rs = 0.40, *p* = 0.04), while helplessness and magnification did not correlate with RGBCC (rs = 0.021, *p* = 0.94 and rs = 0.04, *p* = 0.41, respectively).

### 3.7. Correlation between RGBCC and HADS Score in the LBP Group

Figure 9 shows the correlations of RGBCC with total HADS score and with the scores of the two subscales of HADS. There were no correlations between RGBCC and total HADS score (rs = 0.13, *p* = 0.52), HADS anxiety score (rs = 0.04, *p* = 0.87), or HADS depression score (rs = 0.17, *p* = 0.49).

## 4. Discussion

This study showed that patients with chronic LBP who have LBP-related psychological problems exhibit facial color changes when asked LBP-related questions more than patients with chronic LBP but without LBP-related psychological problems. A video camera was used to detect facial color changes. The change in complexion depends on the blood flow state of facial skin blood vessels. Facial blushing or flushing can be understood as the result of dilation of cutaneous blood vessels of the face. This vasodilation is due to signals from the hypothalamus, which controls stress response, to autonomic nerves [26]. Vasodilation increases blood flow to the skin, making the skin appear reddish. In other words, facial flushing under psychological stress reflects a brain reaction to the stress. Shimo et al. used fMRI to investigate the brain activity of patients with chronic LBP. They demonstrated activation of the cortical area associated with pain and negative emotions after patients saw a picture of a man carrying some luggage in a half-crouching position. They concluded that the virtual LBP stimulus caused memory retrieval of unpleasant experiences that may be associated with prolonged chronic LBP conditions [27]. According to fMRI studies, the amygdala is activated when individuals complain of spontaneous joint pain rather than joint pain caused by external stimuli [28]. We hypothesized that hearing LBP-related questions activates emotional arousal in patients with LBP and triggers changes in their complexion. Using a video camera, we recorded facial images of subjects as they were listening to LBP-related questions, and we converted their facial color to RGB data to detect changes in complexion. The method used to convert complexion into RGB data for analysis was that used in previous studies [29,30]. Our results showed a positive correlation between RGBCC and the score of the rumination subscale of the PCS in the LBP group. Rumination is repeated thinking about the same thing and change in the facial color of a patient when asked an LBP-related question suggests that the patient usually repeatedly remembers the LBP. In an fMRI study of patients with chronic LBP, patients with strong functional imaging of white matter between the dorsomedial prefrontal cortex, amygdala, and nucleus accumbens were found to have a weakened cerebral pain suppression system, with the pain more likely to become chronic; moreover, patients with LBP that persisted for more than one year tended to have a stronger affective factor in the McGill Pain Questionnaire, which is used to detect the negative emotions of patients [6]. A systematic review of the relationship between ruminations and brain activity indicates that the dorsomedial prefrontal cortex plays a role in brain activity during ruminations, and the high activity of the dorsomedial prefrontal cortex in patients with chronic lower back pain may indicate a high role of ruminations in this process [31]. In the present study, cases of chronic lower back pain with high ruminant scores are correlated with facial color because subjects recall negative memories when they hear topics related to lower back pain, they ruminate on memories related to lower back pain, and signals to the autonomic nervous system from the hypothalamus that inhibit the mental stress caused by ruminating lead to vasodilation, and This signal to the autonomic nervous system from the hypothalamus, which inhibits mental stress caused by ruminating, leads to vasodilation and changes in facial color. The change in facial color in patients with chronic lower back pain is thought to occur in patients who are constantly aware of their chronic lower back pain and feel stress. In contrast, even in patients with chronic LBP, significant facial color changes were not observed in patients with low PCS score, and the low PCS score had no relation to the Visual Analog Scale score. Therefore, the facial color change may be limited to patients with negative emotions regarding LBP. Emotions are generally controlled by specific activation of the hippocampus and amygdala. The changes in complexion observed in this study may be associated with activation of the hippocampus and amygdala. It is necessary to conduct further studies to better understand the association between facial color changes and brain activity. Patients with high catastrophic thinking have advanced physical symptoms, even in cases of lumbar spinal canal stenosis, and these patients may need mental support [32]. It is our opinion that such a simple method of detecting emotional changes in patients with chronic LBP will be useful in clinical settings. This study has some limitations. First, the sample size was small. Second, there was gender variability between the LBP group and the control group. Third, the changes observed in the facial images were regarded as reflective of facial blood flow. The results of this study are a very limited view and only yield a relationship with one subscale of the PCS scale. We consider this a significant result, but further research is needed to determine if complexion is solely dictated by this scale. Additionally, various factors other than brain activity may also affect facial blood flow. Combining evaluation using the non-contact brain activity detection sensor used in this study and evaluation using fMRI in future studies may allow for the identification of the brain area associated with the change in facial color observed in this study. 

## 5. Conclusions

In conclusion, in this study, we identified patients with chronic LBP and catastrophic thinking, especially those with high rumination scores, by observing facial images for changes in complexion that occur when the patients hear LBP-related questions. Facial image analysis of chronic lower back pain patients using a non-contact sensor was able to detect association between facial color and psychological state of patients with LBP.

## Figures and Tables

**Figure 1 medicina-58-01471-f001:**
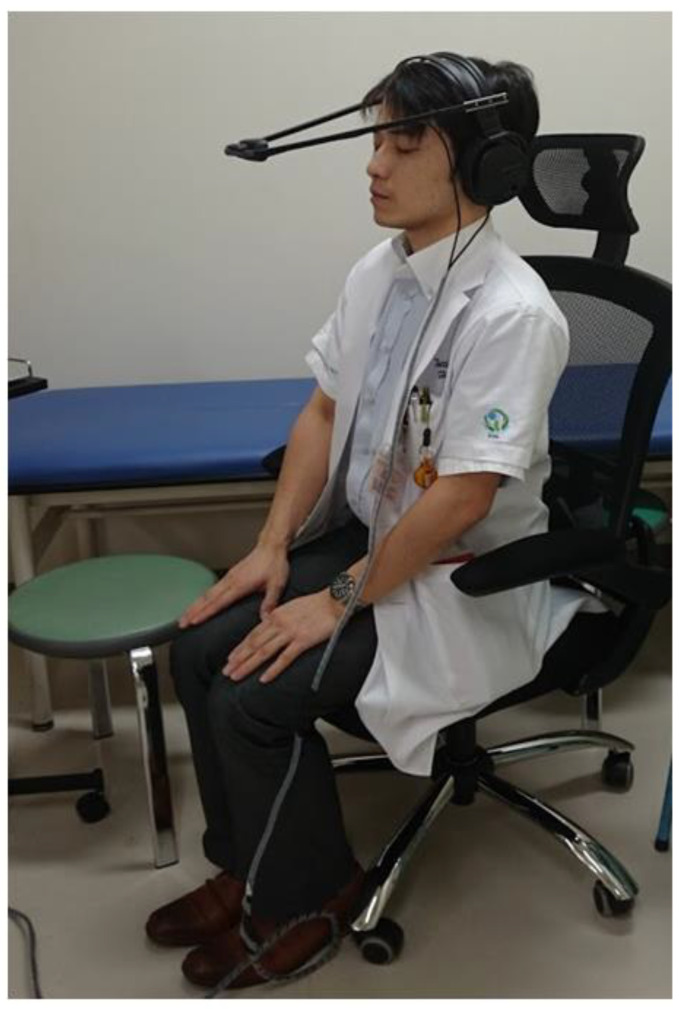
Position in which subjects listened to questions related to lower back pain and general questions unrelated to lower back pain using headphones. This subject consented to the use of the image.

**Figure 2 medicina-58-01471-f002:**
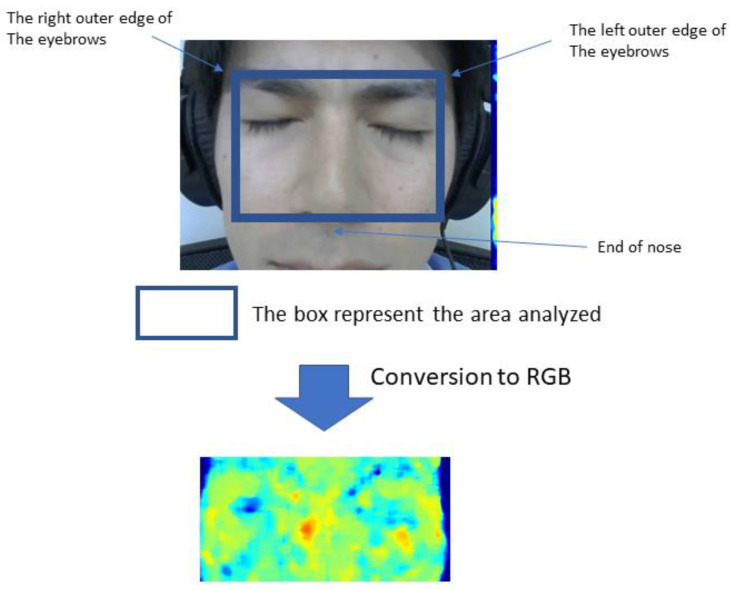
Representative facial image. The region of interest is defined by the outer edge of the eyebrows and the base of the nose. The obtained signal was converted to red, green, and blue values (8-bit color depth) contained in each pixel in the designated area of all image frames. This subject consented to the use of these images.

**Figure 3 medicina-58-01471-f003:**
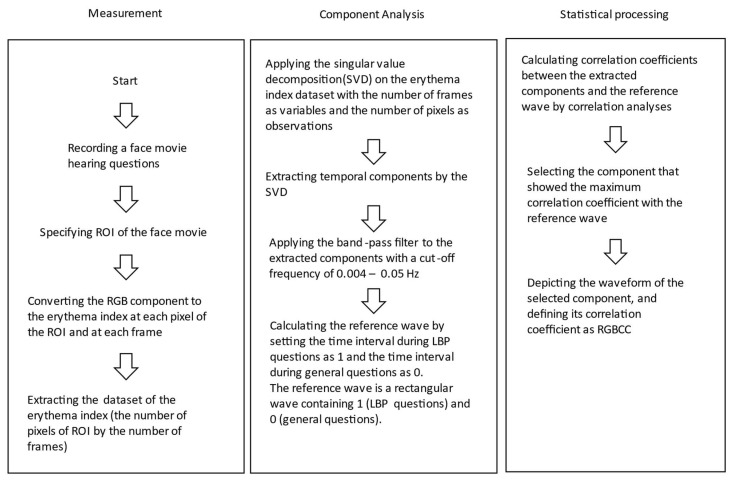
Data processing flow chart for calculating the facial erythema index. MATLAB, which is a multi-paradigm numerical computing environment and proprietary programming language developed by MathWorks, was used. RGB, red, green, and blue; RGBCC, red, green, and blue correlation coefficient; ROI, region of interest. LBP, lower back pain.

**Figure 4 medicina-58-01471-f004:**
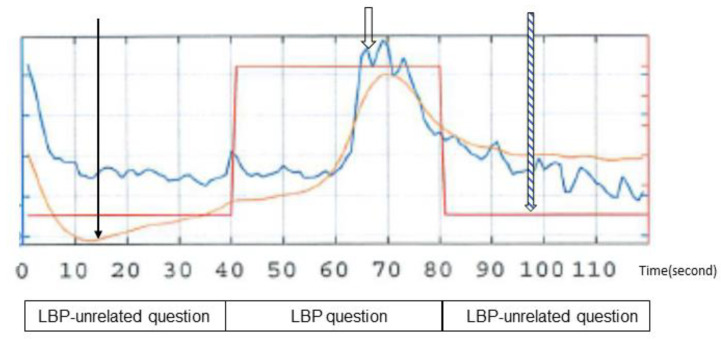
Typical waveforms obtained using the non-contact brain activity detection sensor. (→): component with band-pass filtering, (
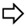
): component without band-pass filtering, (
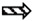
): regressor (wave showing questions related to lower back pain and questions unrelated to lower back pain-).

**Figure 5 medicina-58-01471-f005:**
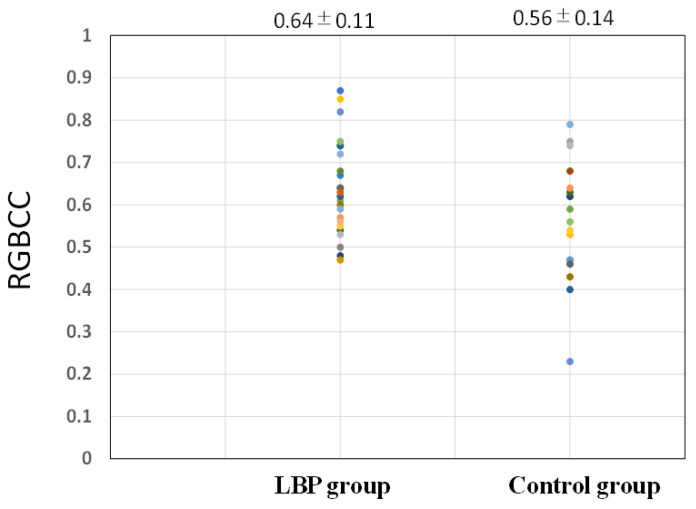
Comparison of red, green, and blue correlation coefficient between the lower back pain group and the control group.

**Figure 6 medicina-58-01471-f006:**
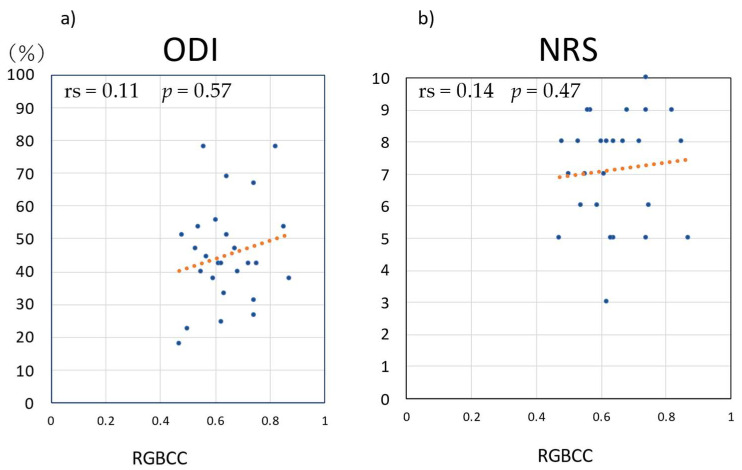
Correlations of red, green, and blue correlation coefficient (RGBCC) with Oswestry Disability Index (ODI) and Numerical Rating Scale (NRS) scores in the lower back pain group. (**a**) RGBCC versus ODI, (**b**) RGBCC versus NRS score.

**Figure 7 medicina-58-01471-f007:**
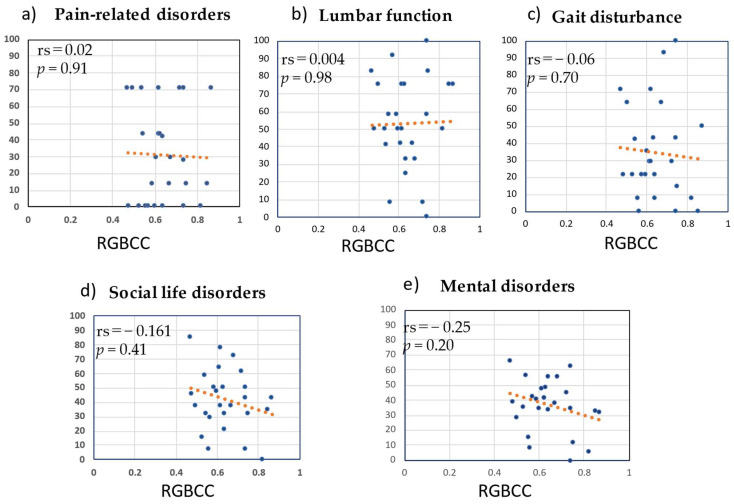
Correlations of red, green, and blue correlation coefficient (RGBCC) with scores of the domains of Japanese Orthopedic Association Back Pain Evaluation Questionnaire. (**a**) RGBCC versus score of pain-related disorders, (**b**) RGBCC versus score of lumbar function, (**c**) RGBCC versus score of gait disturbance, (**d**) RGBCC versus score of social life disorders, (**e**) RGBCC versus score of mental disorders.

**Figure 8 medicina-58-01471-f008:**
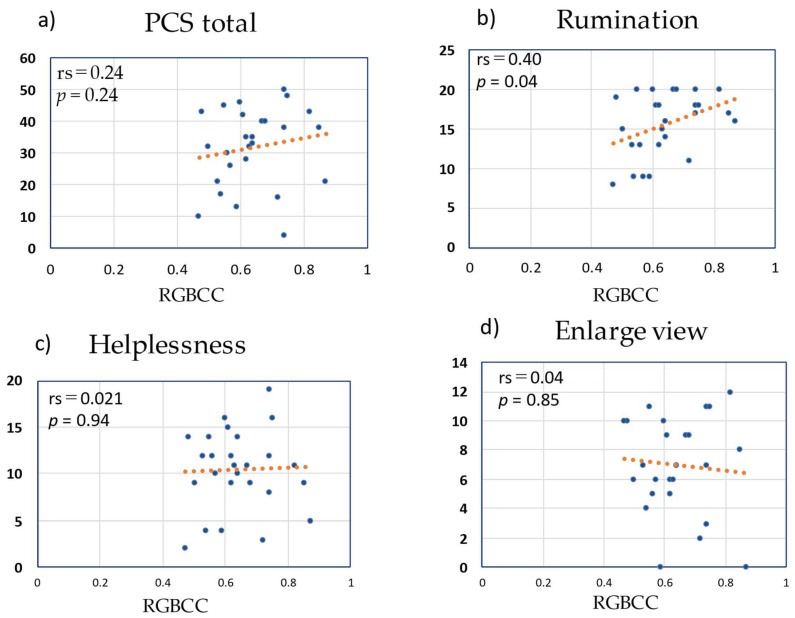
Correlations of red, green, and blue correlation coefficient (RGBCC) with Pain Catastrophizing Scale (PCS) scores. (**a**) RGBCC versus total PCS score, (**b**) RGBCC versus rumination score, (**c**) RGBCC versus helplessness score, (**d**) RGBCC versus magnification score.

**Figure 9 medicina-58-01471-f009:**
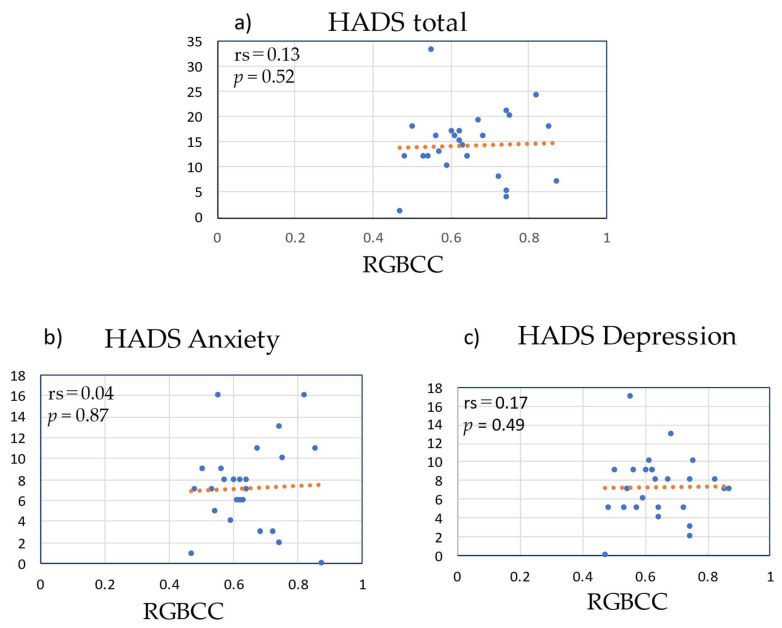
Correlations of red, green, and blue correlation coefficient (RGBCC) with Hospital Anxiety and Depression Scale (HADS) scores. (**a**) RGBCC versus total HADS score, *(***b**) RGBCC versus score of HADS anxiety subscale, (**c**) RGBCC versus score of HADS depression subscale.

**Table 1 medicina-58-01471-t001:** Questions for study subjects.

General questions unrelated to lower back pain	Are you over 20 years of age?
	What part of Japan are you from?
	Do you have siblings?
	What color(s) do you like?
Questions related to lower back pain	What would you like to have (or do) if you did not have lower back pain?
	Would you like to take medications to relieve the lower back pain?
	What do you think is the prognosis of the lower back pain?
	In your opinion, how did your doctor’s approach to diagnosis and treatment affect your lower back pain?
General questions unrelated to lower back pain	Do you like reading books?
	Do you like apples?
	Is the weather good today?
	Is it a hot day today?

**Table 2 medicina-58-01471-t002:** Demographics and clinical characteristics of chronic lower back pain patients.

Characteristic (*n* = 26)	Value *
Age (mean, SD)	68.0 (13.9)
Gender (Male: Female)	9:17
Diseases causing back pain	
Lumbar spinal stenosis	13
Lumbar spondylosis	2
Lumbar vertebral fracture	2
Lumbar kyphoscoliosis	2
Lumbar spondylosis deformans	2
Sacroiliac arthritis	2
Lumbar disc herniation	3
Cases with previous lumbar spine surgery	12
Disease period (month)	43.5 (35.4)
RGBCC	0.64 (0.11)
ODI	45.1 (15.3)
JOABPEQ	
Lower back pain	31.2 (28.7)
Lumbar function	53.4 (26.0)
Walking ability	34.7 (28.2)
Social life function	1.7 (21.4)
Mental health	37.0 (17.2)
PCS (Total score)	28.5 (15.9)
Ruminations	13.1 (6.8)
Helplessness	9.4 (5.1)
Magnification	6.0 (4.0)
HADS (Total score)	12.3 (8.3)
Anxiety	6.2 (4.7)
Depression	5.9 (4.1)
Lumbar NRS (at study)	7.1 (1.7)

* All values are presented as raw numbers and percentages. Oswestry Disability Index (ODI), The Japanese Orthopaedic Association Back Pain Evaluation Questionnaire (JOABPEQ), Pain Catastrophizing Scale (PCS), Hospital Anxiety and Depression Scale (HADS), Numerical Rating Scale (NRS).

**Table 3 medicina-58-01471-t003:** Demographics and clinical characteristics of control subjects.

Characteristic	Value *
Age (mean, SD)	60.8 (16.1)
Gender (Male:Female)	12:6
Diseases	
Knee osteoarthritis	2
Shoulder osteoarthritis	1
Cervical spondylotic myelopathy	2
Rotator cuff injury	1
Hip osteoarthritis	2
Arteriosclerosis obliterans	1
Bruise on the lower leg	1
No disease (healthy subjects)	8
RGBCC	0.56 (0.14)

* All values are presented as raw numbers and percentages.

## Data Availability

Not applicable.

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
