# Peer review of "Relationship between Facial Color Changes and Psychological Problems Associated with Lower Back Pain"

_medicina, 2022, doi:10.3390/medicina58101471_

Round 1

Reviewer 1 Report

Medicina-1937185 REVIEW COMMENTS

COMMENTS

Thanks to the authors for their submission to Medicine. This manuscript investigated whether a non-contact brain activity detection sensor can be used to detect the facial colour of patients with chronic low back pain. Furthermore, it analysed the association between facial colour, low back pain assessment scores and the psychological state of patients with low back pain. I fully acknowledge the time and effort spent in the analysis of the results and the subsequent preparation of a manuscript. However, I consider the results to be inconclusive for the implications described.

SUMMARY

The authors have summarised the paper according to the requirements and in an order that facilitates the understanding of the work.

INTRODUCTION

This section is well written, concise and includes all the information necessary to contextualise and justify the need for this study.

RESULTS

Information on participants, measures and scales are not shown in a results table. It would be interesting to present a table of results differentiating age, sex, pathologies associated with low back pain, RGB and the different scales.

DISCUSSION

The authors did not discuss in detail the results and their interpretation in relation to previous studies and working hypotheses.

CONCLUSIONS

The conclusion section should be clear and concise, and should respond to the proposed objective. I suggest avoiding "The results suggest ...". Also, the possible implications of the results obtained make more sense in the Discussion section. Please consider this and modify this section accordingly.

My personal opinion is that the research is important for the collection of data on the RGB and the different rating scales. However, the results of the study report a very limited view and only obtain a relation to one subscale of the PCS scale. Consequently, the implications of the research are not conclusive to indicate that there is a relationship between psychological problems, low back pain and facial colour change.

Best regards,

Reviewer 2 Report

It is a curious study, and it is technically well elaborated.

I wonder if there is the possibility of a statistical alternative.

otherwise I find it interesting to use measurement alternatives.

Round 2

Reviewer 1 Report

Thank you for the opportunity to review the revised version of this paper. Overall, the authors have sufficiently addressed my concerns. I appreciate the revisions to improve the manuscript and believe the current version is a great improvement over the previous one.